# Whole-Exome Sequencing Identifies Homozygote Nonsense Variants in *LMOD2* Gene Causing Infantile Dilated Cardiomyopathy

**DOI:** 10.3390/cells12111455

**Published:** 2023-05-23

**Authors:** Reiri Sono, Tania M. Larrinaga, Alden Huang, Frank Makhlouf, Xuedong Kang, Jonathan Su, Ryan Lau, Valerie A. Arboleda, Reshma Biniwale, Gregory A. Fishbein, Negar Khanlou, Ming-Sing Si, Gary M. Satou, Nancy Halnon, Glen S. Van Arsdell, Carol C. Gregorio, Stanly Nelson, Marlin Touma

**Affiliations:** 1Department of Pathology and Laboratory Medicine, David Geffen School of Medicine, University of California, Los Angeles, CA 90095, USA; 2Department of Cellular and Molecular Medicine and Sarver Molecular Cardiovascular Research Program, The University of Arizona, Tucson, AZ 85721, USA; tmlarrinaga@arizona.edu (T.M.L.); gregorio@arizona.edu (C.C.G.); 3Neonatal Congenital Heart Laboratory, Department of Pediatrics, David Geffen School of Medicine, University of California, Los Angeles, CA 90095, USA; 4Department of Pediatrics, David Geffen School of Medicine, University of California, Los Angeles, CA 90095, USA; 5Department of Human Genetics, David Geffen School of Medicine, University of California, Los Angeles, CA 90095, USA; 6Molecular Biology Institute, University of California, Los Angeles, CA 90095, USA; 7Eli and Edyth Broad Stem Cell Research Center, University of California, Los Angeles, CA 90095, USA; 8Department of Surgery, David Geffen School of Medicine, University of California, Los Angeles, CA 90095, USA; 9Department of Medicine and Cardiovascular Research Institute, Icahn School of Medicine at Mount Sinai, New York, NY 10029, USA; 10Children’s Discovery and Innovation Institute, University of California, Los Angeles, CA 90095, USA; 11Cardiovascular Research Laboratories, David Geffen School of Medicine, University of California, Los Angeles, CA 90095, USA

**Keywords:** *LMOD2*, leiomodins, heart maturation, thin filament, sarcomere, whole-exome sequencing, whole-genome sequencing, neonatal cardiomyopathy, DCM

## Abstract

As an essential component of the sarcomere, actin thin filament stems from the Z-disk extend toward the middle of the sarcomere and overlaps with myosin thick filaments. Elongation of the cardiac thin filament is essential for normal sarcomere maturation and heart function. This process is regulated by the actin-binding proteins Leiomodins (LMODs), among which *LMOD2* has recently been identified as a key regulator of thin filament elongation to reach a mature length. Few reports have implicated homozygous loss of function variants of *LMOD2* in neonatal dilated cardiomyopathy (DCM) associated with thin filament shortening. We present the fifth case of DCM due to biallelic variants in the *LMOD2* gene and the second case with the c.1193G>A (p.W398*) nonsense variant identified by whole-exome sequencing. The proband is a 4-month male infant of Hispanic descent with advanced heart failure. Consistent with previous reports, a myocardial biopsy exhibited remarkably short thin filaments. However, compared to other cases of identical or similar biallelic variants, the patient presented here has an unusually late onset of cardiomyopathy during infancy. Herein, we present the phenotypic and histological features of this variant, confirm the pathogenic impact on protein expression and sarcomere structure, and discuss the current knowledge of *LMOD2*-related cardiomyopathy.

## 1. Introduction

Neonatal and early infantile cardiomyopathies are rare but carry high mortality rates, mitigated by supportive therapy and cardiac transplantation. Common pathophysiological categories include mitochondriopathies, glycogen storage diseases, arrhythmogenic channelopathies, and defects in structural elements [1,2]. Diagnostic workups consist of electrocardiography, echocardiography, histopathology, electron microscopy, and biochemical testing [2], which normally take place before requesting genetic testing. The genetic workup commonly involves standard cytogenetics [3] and targeted next-generation sequencing (NGS). To date, ClinVar lists 1911 pathogenic or likely pathogenic (P/LP) variants and 2615 variants of uncertain significance (VUS) related to the keyword “neonatal cardiomyopathies” of a size detectable by sequencing (<50 base pairs) with known submission dates. Likewise, it lists 203 P/LP variants and 96 VUS related to “infantile dilated cardiomyopathy”. New variants in each class have been discovered at an accelerating rate each year, with 868 P/LP variants and 993 VUS for neonatal DCM and 79 P/LP and 34 VUS for infantile DCM in the year 2021 alone (Figure 1A,B). These are conservative calculations as many clinically relevant variants are not explicitly labeled with developmental stages for reasons including insufficient cases or variable phenotypic timelines. *LMOD2* (OMIM 608006), which encodes leiomodin-2, is one such gene with three pathogenic variants, listed in May 2022, during the time frame of the current manuscript preparation, manifesting with infantile DCM at variable ages of onset according to recent case reports.

Leiomodins are members of the tropomodulin superfamily that bind the pointed end of actin thin filaments, including smooth muscle leiomodin-1 (Lmod1), cardiac predominant leiomodin-2 (Lmod2), and muscle-predominant leiomodin-3 (Lmod3), which has recently been implicated in congenital nemaline myopathy. Encoded by *LMOD2,* Lmod2 is expressed exclusively in striated muscle and represents the predominant leiomodin in cardiac muscle [4,5]. It has been shown that Lmod2 is essential for the cardiac thin filament elongation to reach a mature length, which in turn is necessary for the proper generation of contractile force [5,6,7]. Defects in thin filament length underlie disease progression in *LMOD2*-KO neonatal mice [4,5] and *LMOD2*-related neonatal cardiomyopathy reported in humans [8,9,10,11]; yet, *LMOD2* has been listed in few cardiomyopathy panels with inconsistent indications for testing, e.g., “familial dilated cardiomyopathy” [12,13], “congenital structural myopathy” [14], and “hypertrophic cardiomyopathy” [15].

A homozygous nonsense variant in *LMOD2* (c.1193G>A, p.Trp398*) causing the absence of the Lmod2 protein, severe shortening of thin filaments in the left ventricle, and contractile force deficiency in patient myocytes has been previously reported in one individual [8]. Here, we present a second case of the same *LMOD2* variant and zygosity as well as a matching DCM phenotype but with the age of onset a few months older than the genetically matching case and other DCM cases with pathogenic *LMOD2* variants. Furthermore, the histological findings are also heterogeneous to some extent. We illustrate the unique clinical and pathological features, investigate the causality, and discuss the current literature evidence to propose the pathobiology of *LMOD2*-related DCM. Finally, we argue for the use of WES/WGS in fetal and neonatal DCM. 

## 2. Materials and Methods

### 2.1. Human Studies

All human studies were conducted in accordance with the regulations of the University of California Los Angeles (UCLA) Institutional Review Board (IRB). All subjects provided written informed consent to participate in this study. Electronic medical records, family pedigree, and specimen collection were acquired through the UCLA Congenital Heart Defect (CHD) BioCore [16] following the UCLA-IRB-approved protocols. All specimens were de-identified and coded following acquisition. DCM diagnosis was determined based on clinical data and pathological examination of the explanted heart. 

### 2.2. Trio Whole-Exome Sequencing (WES)

The genomic DNA of the proband was extracted from peripheral blood monocytes (PBMCs) at the UCLA Congenital Heart Defects BioCore by using standard methods (Purelink Genomic DNA Mini Kit, Invitrogen, Waltham, MA, USA). Library preparation, sequencing, and data analysis were performed at the CCRD (California Center for Rare Disease) and the UCLA Clinical Genomics Center, using the CLIA (Clinical Laboratory Improvement Amendments)- and CAP (College of American Pathologists)-validated protocols. Genomic DNA (3 µg) samples from the proband and parents were subjected to library preparation and exome capture following the Agilent Sure Select Human All Exon 50 Mb Illumina Paired-End Sequencing Library Prep Protocol. Sequencing was performed on an Illumina HiSeq4000 as a 50 bp paired-end run. For each sample, approximately 200 million independent paired reads were generated for an average coverage of 140× of RefSeq protein-coding exons and flanking introns (+2 bp), with at least 95% of these bases covered at ≥10×. The sequences were aligned to the hg19/b37 genome release by using the Novoalign function. PCR duplicates were marked using Picard. The Genome Analysis Toolkit (GATK) [17] was used for indel realignment and base quality recalibration. Both SNVs (single nucleotide variants) and small INDELs (insertions and deletions) were called using GATK unified genotyper. All variants were annotated using the customized VEP (variant effect predictor) engine from Ensembl. Regions of homozygosity by descent were determined using PLINK. Rare variants with a minor allele frequency of <1% in public databases were retained for further analysis.

### 2.3. Variants Pathogenicity Analysis

Candidate rare variants were classified based on their pattern of inheritance/zygosity, their location within the gene, conservation scores, population and allele frequencies based on ClinVar, predicted consequence at the protein level and structural domains, pathogenicity prediction in silico tools, evidence from functional studies and animal models, and disease spectrum. All variants were interpreted in the context of the patient’s phenotype. Variants were dismissed if they were predicted to be tolerant (have low impact on protein structure or function) or have been reported in the GnomAD database. Finally, the technical quality of the candidate variants was confirmed using the Integrative Genomics Viewer (IGV) [18].

### 2.4. Heart Tissue Specimens

Heart tissue sections were obtained through the UCLA CHD BioCore [16]. Control specimens were obtained from a heart transplant donor who died of a noncardiac cause but was noncompatible for transplant. All samples were de-identified and coded upon procurement. Cryosections and glutaraldehyde-fixed specimens for electron microscopy were prepared at the UCLA clinical pathology laboratory.

### 2.5. Histology and Immunofluorescence Staining

For histological analysis, heart tissue specimens obtained in diastolic arrest were fixed in 4% (*v/v*) formaldehyde, embedded into paraffin, and cut into 5 μm thick tissue sections through the clinical pathology laboratory. For immunofluorescence staining, a small LV piece from the proband’s explanted heart, a non-failing control, and the first patient identified with [8] was thawed, stretched, and fixed in 4% paraformaldehyde in 1× PBS overnight. The following day, the tissue was washed 4 times with 1× PBS every hour. The LV sections were embedded in the Tissue-Tek O.C.T. compound (Sakura Finetek) and frozen in 2-methyl butane cooled in liquid nitrogen. Cryosections of 5 μm length were collected on 1.5 size coverslips (that had been previously coated with gelatin), permeabilized with 5 mL of 0.2% Triton-X for 30 min at room temperature and blocked with 50 μL of blocking solution (2% BSA, 1% normal donkey serum in PBS) for 1 h at room temperature. Primary antibodies were applied for 2 h at room temperature and then washed with 5 mL of 1× PBS three times every ten minutes. Secondary antibodies and phalloidin were applied for 2 h at room temperature and then washed with 5 mL of 1× PBS three times every ten minutes. Sections were then rinsed in dH_2_O and mounted onto microscope slides with Aqua Poly/Mount (Polysciences Inc., Warrington, PA, USA). Primary antibodies include mouse monoclonal anti–α-actinin (1:200 EA-53, Sigma-Aldrich, St. Louis, MO, USA) and rabbit polyclonal anti-Tmod1 (12.5 μg/mL) antibodies. Secondary antibodies include Alexa Fluor 488–conjugated donkey anti-rabbit IgG (1:200) and Alexa Fluor 350–conjugated goat anti-mouse IgG (1:200). Texas Red–conjugated phalloidin (1:1000) was used to stain F-actin (ThermoFisher, Waltham, MA, USA). For immunofluorescence staining, well-characterized left ventricular tissue from a healthy donor heart as close in age (14 months) as possible to the patient was obtained from the Sydney Heart Bank [19] (University of Sydney, Camperdown, NSW, Australia) following approval of the Human Research Ethics Committee. The tissue was transported and stored in liquid nitrogen. 

Antibodies Information: Monoclonal Anti-α-Actinin (Sarcomeric) antibody produced in mouse, clone EA-53, from Sigma Aldrich, catalog no. A7811. The Tmod1 antibody was made in-house. There were also Donkey anti-Rabbit IgG (H+L) Highly Cross-Adsorbed Secondary Antibody, Alexa Fluor™ Plus 488 from ThermoFisher, catalog no. A32790, and Goat anti-Mouse IgG (H+L) Highly Cross-Adsorbed Secondary Antibody, Alexa Fluor™ 350, from ThermoFisher, catalog no. A-21049.

### 2.6. Western Blot Analysis

After protein quantification, protein samples were electrophoresed in 4–12% SDS polyacrylamide gels before transfer to PVDF membranes (MilliporeSigma, Burlington, MA, USA). Conjugated secondary antibodies (Bio-Rad, Hercules, CA, USA) were used followed by an ECL reaction to develop the blots according to the manufacturer’s instructions. Band intensities from the film were analyzed by IMAGEQUANT 5.2 software (Molecular Dynamics) if needed. For Lmod2 expression, AP52508PU-N (OriGene, Rockville, MD, USA) was used with Gapdh (Cell signaling, Danvers, MA, USA, catalog no. 2118S) as a loading control.

### 2.7. Thin Filament Length Measurements

A total of 16–21 images (per group) of LV sections with visible Tmod1 doublet staining at thin filament pointed ends were taken using a Deltavision RT system (GE Healthcare, Chicago, IL, USA) with a 100× NA 1.3 objective and a CCD camera (CoolSNAP HQ; Teledyne Photometrics, Tucson, AZ, USA), deconvolved using SoftWoRx software and processed using ImageJ (National Institutes of Health, Bethesda, MD, USA). Thin filament lengths (at least 4 thin filaments per image) were measured by Tmod1 staining using the ImageJ plugin DDecon [20]. 

### 2.8. Statistical Analysis

Quantified results are presented as mean ± SD. Student’s *t*-test (unpaired, 2-tailed) and one-way ANOVA with Tukey’s multiple comparisons test was used for comparing two groups and more than two groups, respectively; a *p*-value less than or equal to 0.05 was considered significant unless specified otherwise. 

## 3. Results

### 3.1. Clinical Presentation

The leading case is a 4-month-old male infant of Hispanic descent who was born at 39 + 6/7 weeks via cesarean section (C/S) due to previous C/S deliveries. At birth, the infant had a normal weight (3.7 kg) and height (50.0 cm). The pregnancy course was uneventful; however, the delivery was complicated by fetal distress, meconium aspiration, and persistent pulmonary hypertension (PPHN) that required hospitalization in the neonatal intensive care unit of an outside hospital (OSH) for approximately one week. The neonatal echocardiogram at 1 day of life showed mild pulmonary artery hypertension demonstrated by a bidirectional shunt through a patent ductus arteriosus and a patent foramen ovale (PFO) versus small secundum ASD (ssASD). The tricuspid and mitral valves showed mild insufficiency. The left and right ventricular functions were both normal. Subsequent echocardiography at one week of life revealed resolution of PPHN features with residual asymptomatic PFO versus ssASD, mild left atrial dilation, and mild left ventricular enlargement, but normal biventricular function was documented. 

At 4 months of age, the infant presented to OSH with dyspnea, pallor, and lethargy, requiring intubation and mechanical ventilatory support. Chest X-rays revealed cardiomegaly. Echocardiography imaging revealed biventricular enlargement and severely diminished biventricular systolic functions, with left ventricular ejection fraction measured at 35% (Simpson’s biplane) and fractional shortening measured at 16% (Figure 2 A,B, Appendix A). The right ventricular function was qualitatively moderately diminished, indicating biventricular dysfunction. The viral upper respiratory panel revealed a parainfluenza infection, while a baseline electrocardiogram (ECG) recorded regular rate and rhythm with a right-deviated axis with QTc 0.315–0.455 s (age-specific reference range: 0.44–0.47) but was not consistent with viral myocarditis as a potential underlying etiology (Figure 2C).

The critical status of the proband prompted the transfer to the author’s institution and admission to the pediatric intensive care unit for further management of acute cardiorespiratory failure. The hospital course was complicated by septic shock due to *Enterococcus* and *Staphylococcus epidermidis*. After an episode of wide-complex tachyarrhythmia on day 11 of admission (Figure 2D), the infant was placed on veno-arterial extracorporeal membrane oxygenation (VA-ECMO) and then transitioned to a Berlin left ventricular assist device (LVAD) on day 15 as a bridge to heart transplant. The echocardiogram demonstrated stabilization following LVAD implantation. Left ventricular ejection fraction and fractional shortening, which showed severe hypomobility before transplantation, each in the range of 15–30% and 5–20%, respectively, normalized immediately after the transplant, each to the range of 60–80% and 30–45%, respectively (Figure 3A,B); electrocardiographs showed no significant change in the ventricular rate (Figure 3C), and both QT and QTc intervals decreased (Figure 3D). Brain natriuretic peptide titers also normalized and stabilized from a state of persistent fluctuation and severe elevation (Figure 3E). Eventually, the infant received an ABO-compatible orthotopic heart transplant (OHT) at 7 months of age. 

The family history did not reveal consanguinity. Both parents and the two female siblings are healthy to date. The mother had eight pregnancies but only three live births, with a history notable for three spontaneous abortions and two fetal demises for unknown reasons at unknown gestational ages. There was no history of heart defects, arrhythmias, or sudden deaths in the family, although the etiology of recurrent abortions and fetal loss was unclear. The three-generation family pedigree is shown in (Figure 4A).

### 3.2. Laboratory Testing

Further extensive workup ruled out other potential causes of cardiac injury and secondary heart failure including hypothyroidism, congenital syphilis, immune hemolytic anemia, and viral infections other than parainfluenza type 4. Routine newborn screening was reportedly normal. Together, the data suggest a working diagnosis of primary infantile cardiomyopathy or a storage disorder (Appendix A).

### 3.3. Cardiac Biopsy

In the present case, a core cardiac biopsy was performed during LVAD placement, and no evidence of myocarditis was observed; however, focal vacuolization of fibers and a mild increase in glycogen deposits were noted on H&E-stained sections (Figure 4(Ba)). Otherwise, nonspecific findings due to fixation were encountered. 

The explant heart, post-implantation of Berlin Excor LVAD, pathology was investigated for correlation. Grossly, the explant showed biventricular dilated cardiomyopathy. The wall thickness of the left ventricle and the right ventricle measured 0.5 cm and 0.2 cm, respectively. Periodic Acid Schiff with and without diastase (PAS/PAS-D) staining did not show a significant increase in glycogen (Figure 4(Bb)). Likewise, Iron staining did not reveal hemosiderosis.

Electron microscopy (EM) panels at direct magnification ranges of ×5000 to ×10,000 showed sarcomeric separation and disruption by excess fluid (edema) and mitochondria. The Z-bands were noted to be disorganized. However, the study resolution did not allow precise evaluation of the thin filament of the myofibrils (Figure 4(Bc)). Reactive mitochondrial swelling and pleoconia were noted. Very rare glycogenosomes were noted, but there is no convincing evidence for excess storage material, organelle abnormality, or abnormal inclusions (Figure 4(Bd)).

### 3.4. Genetic Workup

#### 3.4.1. Molecular Genetics Panels

Comprehensive panels for cardiomyopathy and glycogen storage disease were requested at an OSH. Their gene lists are provided in Appendix A. The cardiomyopathy panel covering 100 known cardiomyopathy genes revealed 2 heterozygous variants of uncertain significance (VUS): *PKP2* c.895C>T (p.R299C) and *SCN5A* c.5872C>T (p.R1958*). Meanwhile, the glycogen storage disease panel covering 29 genes showed a heterozygous VUS in *SLC37A4*, c.1176T>G (p.S392R). *LMOD2* is not included in these panels.

#### 3.4.2. Whole Exome Sequencing (WES)

A trio WES was conducted using peripheral blood monocytes (PPMCs)-isolated gDNA from the proband and both parents. There were 4 regions of homozygosity totaling 6.5–9.2 Mb, spanning less than 1% of the genome. No region of homozygous deletions contained clinically significant genes. No variants were qualified as pathogenic or likely pathogenic according to the American College of Medical Genetics (ACMG) guideline [21]. The previously reported variants in *PKP2*, *SCN5A*, and *SLC37A4* were present, each inherited from the father, father, and mother, respectively. In addition, *CTNNA3* c.293-1G>C was present in a heterozygous state in the proband and the father and was deemed not reportable. Lastly, a biallelic homozygous *LMOD2* c.1193G>A (p.W398*) variant was identified in the proband with both parents heterozygous (Figure 5A,B).

#### 3.4.3. Interpretation of Candidate Variants

##### Correlation of WES Results with Findings from Targeted Molecular Genetics Panels 

The previously reported variants in *PKP2*, *SCN5A*, and *SLC37A4* were also found in WES data, each inherited from the father, father, and mother, respectively. These three variants are known to be associated with autosomal dominant diseases with partial phenotypic overlaps but ultimately remained as VUS since heterozygotes exist in the healthy population and the heterozygous states were established in the healthy parents. Therefore, all three variants were excluded based on the evaluation of phenotypic concordance, population frequency, inheritance mode, and current literature evidence.

##### Identifying New Candidate Genetic Variants via WES

***CTNNA3* VUS.** An additional VUS absent from the targeted panels was detected by WES. The *CTNNA3* gene codes alpha-T-catenin, an 895-amino acid cell adhesion protein that binds with plakophilins and functions in the cell–cell adhesion in cardiomyocytes [23]. It is predicted to tolerate the heterozygous loss of function (pLI < 0.00001 in ExAC). Pathogenic variants of this gene are associated with familial ARVC Type 13 (ARVC13). According to Splice AI, the c.293-1G>C canonical splice site variant is located one base 5′-ward to the first codon of exon 4 and is expected to result in a splice acceptor loss in the middle of the codons for the 98th amino acid. GnomAD reports one heterozygote with this variant. Together, the absence of clinical correlates, the lack of prior loss-of-function variants associated with ARVC13 in a gene that is tolerant to loss of function, and the existence of heterozygotes in a population database and the healthy father rendered the significance of this variant uncertain.

***LMOD2* Homozygous Variant**. The *LMOD2* gene is located on Chromosome 7, spanning three exons. It codes leiomodin-2, a 547-amino acid actin-binding protein involved in actin nucleation, a process that involves the formation of an actin nucleus that serves as the starting point of actin polymerization. Lmod2 is made of several well-conserved functional domains including a tropomyosin-binding domain (TM-h) near its N-terminus, three actin subunit-binding domains (A-h, LRR, and W), proline-rich domain (poly-P), two helical domains (h1 and h2), and a basic amino acid-rich domain (B) (Figure 5C). The proline-rich motifs are ubiquitous and are thought to facilitate transient protein–protein interactions [24]. At the functional level, Lmod2 binds to actin polymers at its pointed end [25], leading to the elongation of actin thin filaments to reach mature lengths within the cardiac sarcomere units [4,5,6,7].

*LMOD2* is predicted to be intolerant of recessive loss of function (pRec = 0.886 in ExAC). One heterozygote and no homozygote has been reported in GnomAD. Nine variants of less than 50 bp lengths have been reported to ClinVar, four of which are pathogenic, as discussed below. Importantly, all reported pathogenic variants to date have a homozygous or compound heterozygous mode of inheritance and are deletion variants. 

Clinical cases and mouse models are available with results summarized in (Table 1). To date, one clinical case of neonatal dilated cardiomyopathy with an identical homozygous variant was reported by Ahrens-Nicklas et al. (2019) [8]. This patient exhibited severe DCM soon after birth requiring LVAD until a cardiac transplant at ten months old. Greenway et al. reported two siblings with compound heterozygous nonsense variants, p.L415fs*108 (c.1243_1244del) and p.R513* (c.1537C>T) [9], the younger of whom had documented cardiac wellness prenatally until a qualitative sign of heart failure in the 37th gestational week. Both siblings had therapeutic withdrawal within 31 days of life due to severe neonatal DCM. The p.L415fs*108 (c.1243_1244del) frameshift variant was also recently reported by Lay et al. in a homozygous state in a nine-month-old patient whose DCM was triggered by a gastrointestinal condition of an unrelated etiology [10]. Lastly, a homozygous canonical splice site variant c.273+1G>A was reported by Yuen et al. in two siblings who had a neonatal-onset DCM and died within 9 and 17 h of life [11].

The c.1193G>A (p.W398*) variant identified in our proband occurs at a moderately well-conserved base of *LMOD2,* affecting the gene product between the LRR domain and proline-rich (poly-P) domain. If this were translated, the encoded protein would lose the poly P, h1, B, h2, and W domains (Figure 5C) [22,24,25,26]. Splice AI did not predict a splicing effect. 

A nonsense variant with a significant loss of amino acids can result in either a truncated protein or a null state by transcription inhibition, nonsense-mediated decay, or misfolding-mediated protein decay. Foregoing clinical cases support nonsense-mediated decay mechanisms. Ahrens-Nicklas et al. showed normal pre-mRNA levels, reduced mature mRNA levels, and absent Lmod2 protein on the Western blot [8]; in Yuen et al., the c.273+1G>A homozygotes showed absent *LMOD2* transcript and protein [11]. In our case, Western blot analysis using an antibody raised against a peptide mapping near the N-terminus of the human Lmod2 (AP52508PU-N, OriGene) revealed that the Lmod2 protein is severely depleted in both ventricles in our proband (Figure 6A) compared to control. The upper faint bands are likely a nonspecific background only. The LMOD2 p. Trp398* mutation encodes a premature stop codon predicted to result in a truncated Lmod2. Therefore, the upper band should not be interpreted as Lmod2. There is possible cross-reactivity with similar epitopes on other proteins. These bands cannot be seen in the control samples because of the strong Lmod2 bands since the Western blot procedure relies only upon the separation of proteins by size using gel electrophoresis. The molecular difference predicts that the lower bands are truncated Lmod2 (c.1193G>A (p.W398*) corresponding to an MW change of 149aa for about 17 kDa. Nevertheless, for more convincing identification, other protein identification methods will be needed to separate the sample as a single protein and digest the protein into peptides, followed by mass spectrometry and bioinformatics analysis. 

At the nanoscale, all available clinical reports and two mouse models of homozygous *LMOD2* deletion [4,27] have reported disorganized myofibrils and short thin filaments under either immunofluorescent staining or electron microscopy. These changes are comparable to the present case (Pt.2) as observed by immunofluorescent staining (Figure 6C, bottom panel). Disorganized myofibrils are exemplified by the observation of Z-line streaming via staining for a Z-disc marker α-actinin and the wide, nonuniform staining of Tmod1, a marker for thin filament pointed ends. Extremely short actin thin filaments are observed in the proband (Pt.2) by staining for filamentous actin (F-actin) and the pointed-end marker Tmod1 (Figure 6C, bottom panel). Tmod1 staining flanks the Z-discs, indicating extremely short thin filaments (0.39 ± 0.09 µm) like what is seen in the first patient (Pt.1) identified with the *LMOD2* c.1193G>A (p.W398*) pathogenic variant (0.29 ± 0.05 µm) [8] (Figure 6B). In comparison, thin filaments in non-failing control tissue are much longer and extend from the Z-discs to the middle of the sarcomere (0.99 ± 0.07) (Figure 6B,C, top panel). Interestingly, Pt. 2′s thin filaments are significantly longer than Pt.1, and sarcomere length is reduced in both patients (Figure 6B).

Microscale histological findings form a broader spectrum. On one extreme, Ahrens-Nicklas’ case showed somewhat disorganized and hypertrophic myocytes with perinuclear clearing, which was also present in the compound heterozygous sibling study [9]. Conversely, the mouse model with PiggyBAC [27] exhibited no specific histological change in cardiomyocytes. The present case has focal vacuolization of fibers and nonspecific reactive changes in mitochondria (Figure 4B).

Electrocardiographic findings appear nonspecific and secondary to physiological stress. Two of the four reports suggest that *LMOD2* may be associated with ventricular arrhythmias. Ahrens-Nicklas reported intermittent episodes of monomorphic ventricular tachycardia and ectopy while on LVAD [8]. Three patients had supraventricular tachycardia [9,10,11]; the piggyBac mouse model showed a prolonged QTc interval [27].

Several pathophysiological interplays of long QTc and heart failure as a cause of arrhythmia were entertained. On one end, genetic long QT syndromes (LQTS) are strongly associated with ventricular arrhythmia, namely, torsade de pointes and T-wave alternans [28]. On the other end, heart failure by itself is a known risk factor of ventricular arrhythmia [29], with multiple induced abnormalities in ion channel activities observed in animal models [30]. Furthermore, the two risk factors of arrhythmia are not completely independent. Primary structural heart diseases, including non-ischemic DCM, are known to raise one or more measurements of the QT interval [31]. Conversely, genetic LQTS patients showed decreased systolic and diastolic functions and increased global longitudinal strain [32]. This study serves to caution against generalizing this relation to other arrhythmogenic genes where the changes in the mechanical parameters are gene-specific. In our case, ECG showed a short QTc. Therefore, a primary mechanical cause of contractility failure is thought to drive arrhythmias in homozygous *LMOD2* non-expressing individuals.

In conclusion, the variant was classified as pathogenic. While it resembled the organ-specific clinical course of prior reports, the discordant disease onset, histology, and electric activities remain to be explained.

## 4. Discussion

The present case of infantile cardiomyopathy is the second report of p.W398* *LMOD2,* the fifth report in the homozygous loss of leiomodin 2 protein expression, and the seventh patient of *LMOD2*-related DCM. This case is significant for confirming the pathogenic impact of this variant and manifesting as a relatively late infantile-onset DCM.

### 4.1. Phenotypic Heterogeneity

The present case and Lay et al. [10] are exceptions to the profound neonatal-onset dilated cardiomyopathy manifested within the first day of life. Both cases presented with a provoked DCM at 4 to 9 months of age, received bridge therapies to an OHT, and had favorable post-transplant outcomes. 

Too few cases are present to contrast the outcomes statistically against the neonatal-onset DCM subset, which had 2 deaths within 1 day of life, 2 therapeutic withdrawals within a month of life, and 1 successful cardiac transplantation at an age comparable to the 2 infantile cases; however, the trend towards a favorable outcome in the infantile cases suggests that age-associated tolerance to physiological stress may be protective once the patient survives the neonatal period.

Specific variants do not fully explain the degree of wellness of the neonatal period, as the two infantile cases both share variants with the neonatal cases. Ahrens-Nicklas demonstrated that a truncated Lmod2 protein, if introduced via cDNA to bypass nonsense-mediated decay, partially rescues the cardiac phenotype in *Lmod2* knockout mice [8]. Therefore, truncated protein expression level may determine the delayed onset and the need for a physiological trigger. While Ahrens-Nicklas’ neonatal-onset case had undetectable protein on Western blot, ours had a faint band, which could explain the late onset, as well as the slightly longer thin filaments in the infantile case (Pt.2). Therefore, based on the results of Western blot analysis and thin filament length measurements, we propose that residual expression of the truncated Lmod2 protein may partially preserve a ventricular function early after birth. 

Regardless of the precise age of onset, the fetal circulation pattern appears to be protective, as cardiac function remains preserved before birth. It was demonstrated that Lmod2 KO mouse heart exhibits thin filament shortening prenatally and before the onset of cardiac symptoms [4]. This observation requires further investigation. Three explanatory hypotheses can be drawn: a pure effect of physiology while gene expression is unchanged, a physiologically induced change in gene expression, or developmental stage-specific gene expression. Yuen et al. hypothesized that this disease does not manifest before birth due to the dramatic physiological hemodynamic changes that occur after birth. As an alternative explanation, they hypothesized that “*LMOD3* might compensate for the loss of *LMOD2* prenatally”. However, given the lack of evidence of any overlapping function and the absence of cardiac phenotypes in Lmod3 KO mice and humans with *LMOD3* mutations [11], this possibility remains to be tested.

Collectively, the postnatal left ventricular changes of the present case (Figure 6A,B) indicate that left ventricular *LMOD2* expression is necessary for normal myocardial maturation in the left ventricle, at least in the early postnatal period. We were not able to measure thin filament length in the RV due to the lack of sufficient appropriate control. However, since Lmod2 is expressed in both ventricles, we propose that Lmod2 is needed for both ventricles. Longitudinal, fetal chamber-specific expression models of *LMOD2* are desired to address these questions. 

### 4.2. Physiological Sarcomerogenesis

Sarcomere develops gradually throughout gestational age. Animal models have demonstrated incomplete cardiac sarcomeres, with Z-bands, actin, and myosin filaments but no M-bands until mid-gestation when interventricular pressure gradually develops and the cardiomyocytes change from round to elongated and spindled with myofilaments aligned under the sarcolemma [33]. When fetal and early neonatal cardiomyocytes undergo mitosis, the sarcomeres are completely disrupted into individual proteins and then reassembled in the daughter cells [34]. The exact timeline of fetal and neonatal *LMOD2* expression and its role in dissembling and reassembling the sarcomere on mitotic versus non-mitotic cardiomyocytes have not been documented but may shed light on when sarcomere disorganization starts, especially whether it precedes physiological effects. Molecular drivers of embryonic cardiomyocyte hyperplasia (proliferation) and postnatal cardiomyocyte hypertrophy differ greatly. The former is positively driven by the *NOTCH*, *WNT*, *NRG-ERBB*, and hypoxia-induced *HIF1* pathways and negatively driven by *HIPPO*, *MEIS1*, and normoxia; the latter is governed by insulin, *IGF1*, *VEGF*, *PI3K-AKT-mTOR*, and *AMPK* pathways [33,34,35]. The pathways that affect *LMOD2* remain to be defined. Selective modulation of any of these pathways or environmental factors with concurrent observation of changes in *LMOD2* expression level may help explain its differential influences on cardiomyocyte phenotype and prenatal versus postnatal circulatory physiology.

In animal models of sarcomerogenesis, ventricular cardiomyocyte stress is known to exert expression changes in selected genes [36,37]. Increased prepartum and immediate postpartum cardiomyocyte proliferation, governed by hypoxic stress, may offset survival advantage after cardiomyocytes lose proliferative capacities. In a mouse model, male, but not female, fetuses carried by mothers kept in a hypoxic chamber demonstrated increased cardiomyocyte progenitor cells [35]. Pending definitive identification of the molecular regulators of *LMOD2* expression as a hypoxia-responsive element, these observations potentially indicate that neonatal hypoxia may keep the myocardial protein expression in “fetal mode”, including *LMOD2* suppression, and thus may prolong the time window of sarcomere disassembly/reassembly in the proliferating neonatal cardiomyocyte. Of note, both the present case and Lay’s case are males, while the neonatal-onset cases are composed of three females and two males. In addition, the present case was complicated by meconium aspiration and respiratory failure, which could contribute to prolonged postnatal hypoxemia. 

A larger population of biallelic *LMOD2*-inactivated neonates and infants, males, and females, with longitudinal documentation of prenatal and perinatal parameters of hypoxic stress plus a concurrent measurement of cell cycle markers is required to translate its significance to human findings. 

### 4.3. WES or WGS as a First-Line Genetic Test for Neonatal and Infantile Cardiomyopathy

Commonly ordered NGS tests for cardiomyopathies include targeted gene panels, WES, or whole-genome sequencing (WGS). The sizes of targeted panels range from 54 by Invitae to 250 by Fulgent according to the National Library of Medicine Genetic Testing Registry NLM GTR, using the search keyword “neonatal dilated cardiomyopathy” and excluding broader indications such as neonatal crisis. Yet, it is rare to find a panel that fits a clinical presentation precisely, particularly the age of onset, even though molecular landscapes of normal cardiac function and dilated cardiomyopathy differ between fetal, neonatal, pediatric, and adult periods [38,39,40].

The increasing size and diversity of panels have raised a question on the consistencyof inclusion criteria [41] in the face of rapid discovery each year [42,43] and the cost- effectiveness compared to WES or WGS. In adult-onset familial DCM, WGS has performed at a high concordance with targeted panels with the advantage of broader and more uniform coverage [44]. For intellectual disabilities, ordering WGS at the onset of genetic workup demonstrated an overall superior diagnostic yield at 37% compared to a secondary WGS following chromosomal microarray (CMA) at 27% [45,46]. WES in conjunction with a targeted panel has shown a 33% diagnostic yield in familial structural heart diseases [47]. Nonetheless, the diagnostic yield and cost effectiveness of WES for neonatal cardiomyopathy remain to be studied. 

Currently, *LMOD2* is inconsistently included in congenital to pediatric cardiomyopathy panels. All authors who reported pathogenic variants in *LMOD2* to date used WES except Greenway et al. 2021 [9], who used a 184-gene panel through Blueprint Genetics (Helsinki, Finland). It is possible to argue that *LMOD2* should be included on the cardiomyopathy panel since six cases of similar genotypes and phenotypes provide solid evidence of its pathogenicity. On the other hand, utilizing WES or WGS as a first-line genetic test would avoid overlooking rare variants, which often lag in their inclusion in targeted panels and may present with yet undocumented phenotypes—such as age-specific categories of DCM. Parsimonious targeted panels used to be appropriate when relatively few genes and variants and their actionability were known for any disease category. Still, WES or WGS as a first-line test has become a reasonable choice [48] given the dropping cost of massively parallel sequencing per case, rapid rate of discovery, and relevant gene therapies on the rise. 

## 5. Conclusions

The present case provides important additional evidence linking biallelic loss-of-function variants of *LMOD2* with profound infantile DCM due to impaired filament elongation and sarcomere maturation during the neonatal and early infantile period. WES allows early detection of *LMOD2* variants during this critical postnatal window, informs clinical decisions and planning, and facilitates family counseling. WES or WGS as a first-line test is a reasonable choice for early diagnosis and management of neonatal and infantile cardiomyopathy.

## Figures and Tables

**Figure 1 cells-12-01455-f001:**
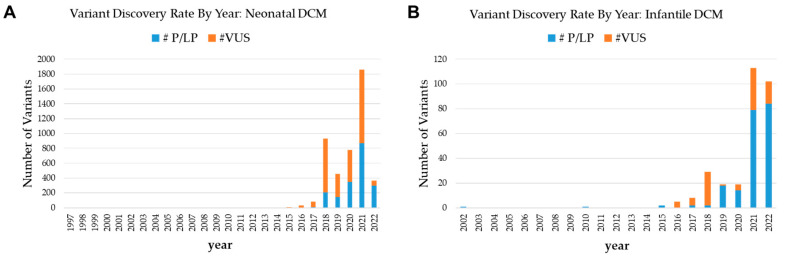
Trends of Variant Discovery Rate in Neonatal and Infantile Dilated Cardiomyopathy (DCM). ClinVar variant discovery by year, filtered with the exact keywords: (**A**) “neonatal dilated cardiomyopathy”, (**B**) “infantile dilated cardiomyopathy”. The number of pathogenic or likely pathogenic (P/LP) variants and variants of uncertain significance (VUS) show a large year-to-year variation with an overall accelerating trend.

**Figure 2 cells-12-01455-f002:**
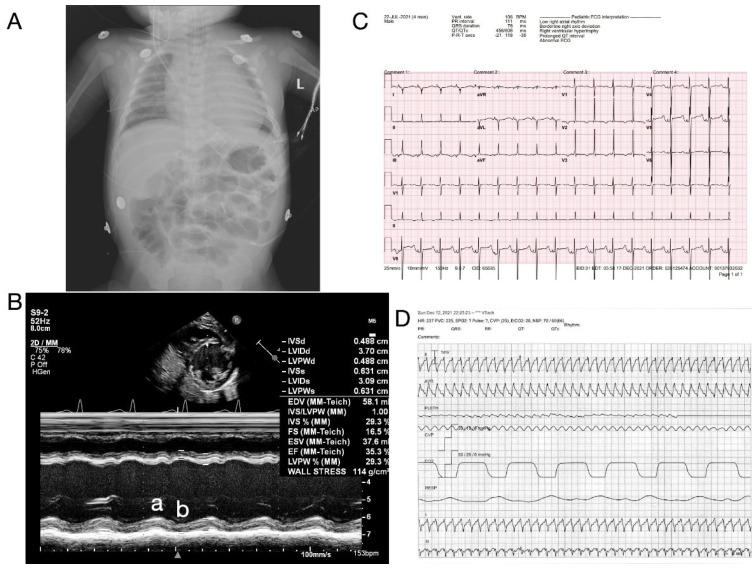
Clinical Presentation of The Proband. (**A**) Chest X-ray on admission demonstrating severe cardiomegaly. L: Left. (**B**) Representative echocardiogram of LV systolic function (M-Mode) at the level of papillary muscles during systole (a) and diastole (b), demonstrating global left ventricular hypokinesis with estimated EF of 35% and fractional shortening (FS) of 16.5% and significant decrease in left ventricle wall thickness. Please see the accompanied Appendix A included in Appendix A. (**C**,**D**) Representative electrocardiogram recording at the baseline (**C**) on admission and during ventricular tachycardia episode (**D**) on day 11 of admission.

**Figure 3 cells-12-01455-f003:**
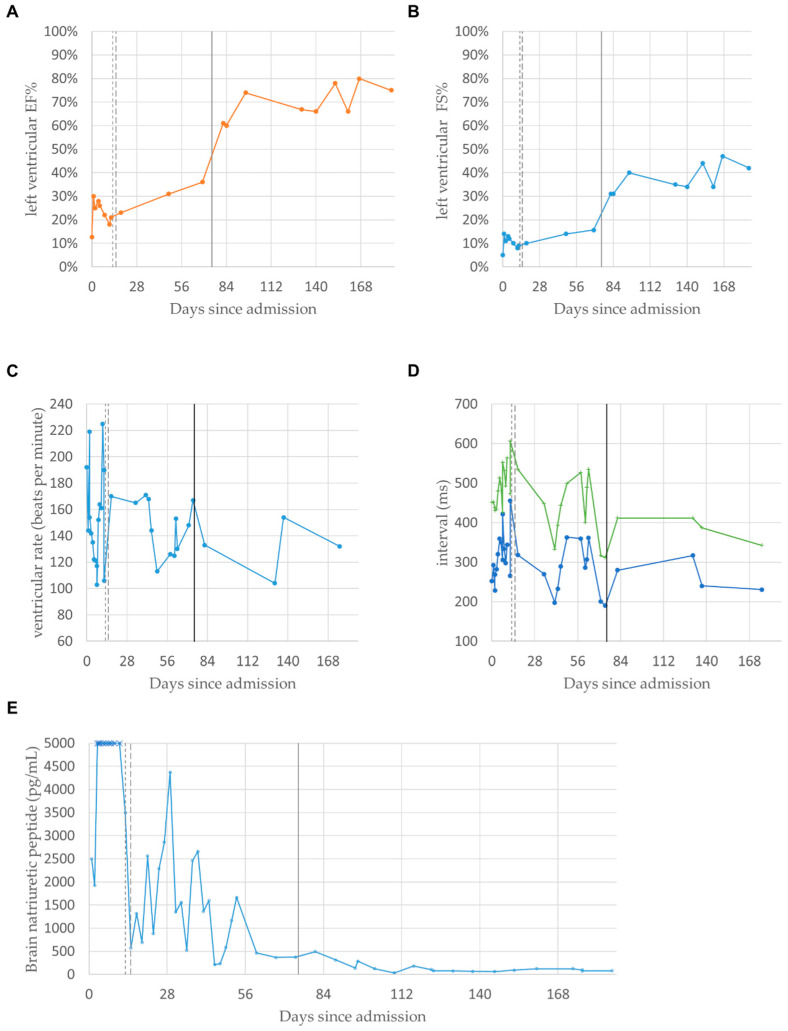
Summary of Cardiac Function Parameters Throughout the Hospital Course. (**A**) Representing graph summary of percentile left ventricle ejection fraction (LVEF%) measured by the unidimensional mode (M-mode). (**B**) Representing graph summary of percentile left ventricle fractional shortening (LVFS%) measured by the unidimensional mode (M-mode). (**C**) Representing graph summary of heart rate. (**D**) Representing graph summary of T I QT nterval (circles) and corrected QT (QTc, crosses), with line break representing the arrhythmia event where these parameters were not measurable. (**E**) Representative graph summary of serum brain-natriuretic peptide (BNP); plateau at 5000 pg/mL represents the upper limit of quantification. For all panels, the vertical lines, from left to right, are dates of placement of ECMO (day 13, fine broken line), LVAD (day 15, coarse broken line), and cardiac transplant (day 75, solid line). As of 11 months of age, the patient has recovered from OHT and sustained gaining weight and height appropriately with occasional mild infections. Echocardiogram has demonstrated stable graft function.

**Figure 4 cells-12-01455-f004:**
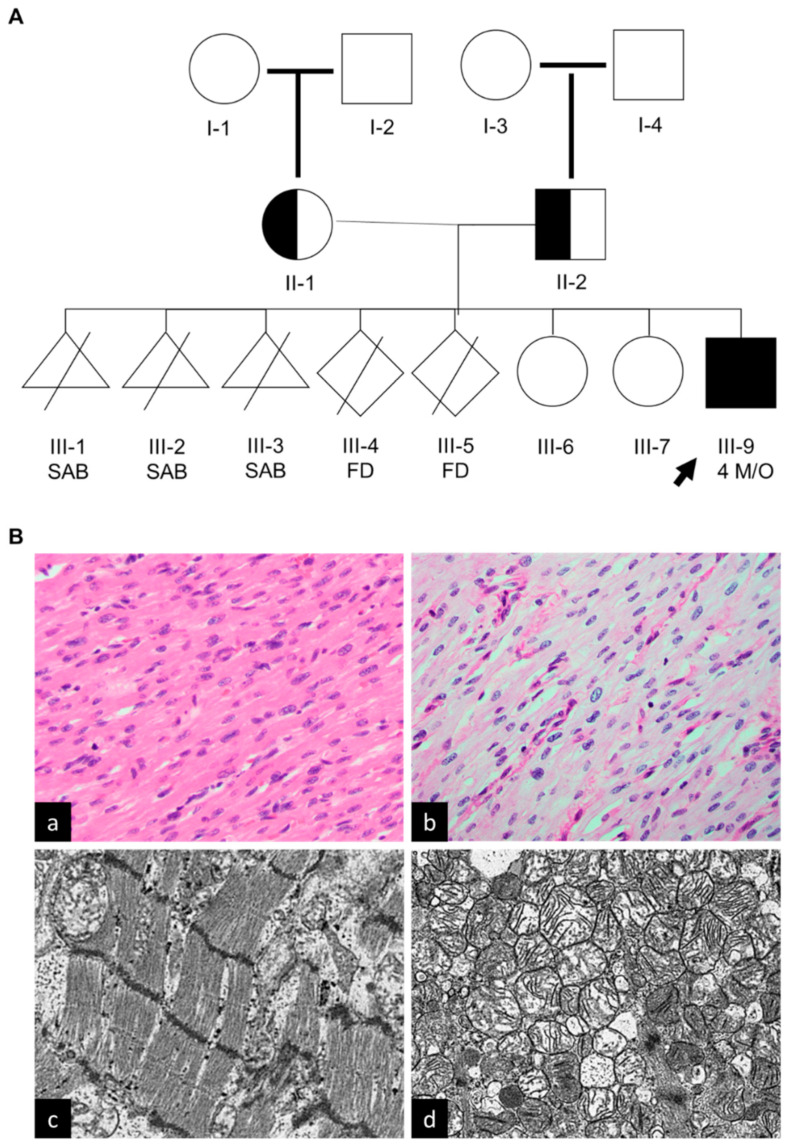
Family Pedigree and Histopathology. (**A**) Three-generation family pedigree: III-9, Index case; II-1, healthy mother carries the c.1193G>A (p. Trp398*) variant; and II-2, healthy father carries the c.1193G>A (p.Trp389*) variant. III-7 and III-8, older healthy sisters that have not been tested. No other family history of heart disease. III-9, male proband (black arrow), presented at 4 months of age with cardiomegaly and profound heart failure. SAB, spontaneous abortion; FD, Fetal Demise. (**B**) Histology of myocardial tissue obtained during left ventricle assist devise implantation (Pre-LVAD). (**a**) Mild cellular vacuolization on light microscopy (H&E stain, 400× magnification); (**b**) Periodic Acid Schiff stain showed mild increase in glycogen (PAS stain, 400× magnification); (**c**) electron microscopy representative imaging (×10,000) depicts disorganized Z-bands, separation of sarcomeres by edema, and rare cytosolic glycogen; no significant lysosome-bound glycogen is noted; (**d**) electron microscopy representative imaging (×50,000) of left ventricular biopsy depicts focal increase in mitochondria with nonspecific mitochondrial swelling and pleoconia noted, with no significant cristae abnormality or inclusions.

**Figure 5 cells-12-01455-f005:**
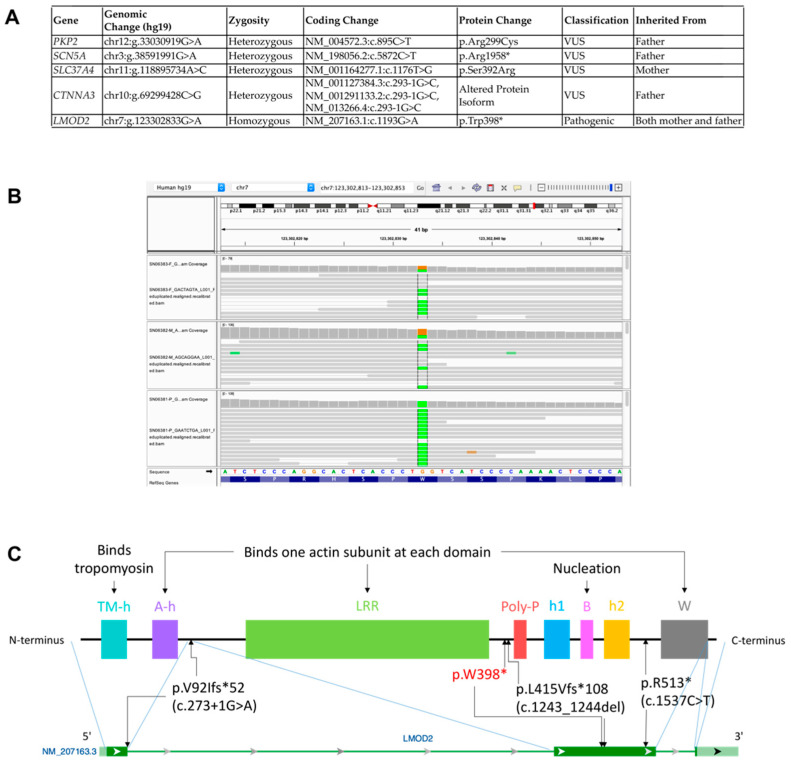
Homozygous Loss of Function Variant of *LMOD2* Revealed by Trio (Proband/Parents) Whole Exome Sequencing (WES). (**A**) Table summary depicts rare heterozygous variants of uncertain significance in *PKP2*, *SCN5A*, and *SLC37A4* genes and homozygous loss of function variant in *LMOD2* detected by WES. (**B**) Integrated genomic viewer (IGV) windows depict homozygous c.1193G>A variant in *LMOD2* in the proband genome, while both parents are heterozygous carriers. Green represents the adenine variant; brown represents guanine base (wildtype). (**C**) ***** Schematic representation of Lmod2 protein with known structural motifs (upper, adopted from Chen et al. 2015 [22]) and *LMOD2* gene (lower). Black arrows indicate exons affected by previously reported deleterious variants. p.W398* indicates the position of the substitution in the present proband caused by the c.1193G>A variant. Please see accompanied Appendix A.

**Figure 6 cells-12-01455-f006:**
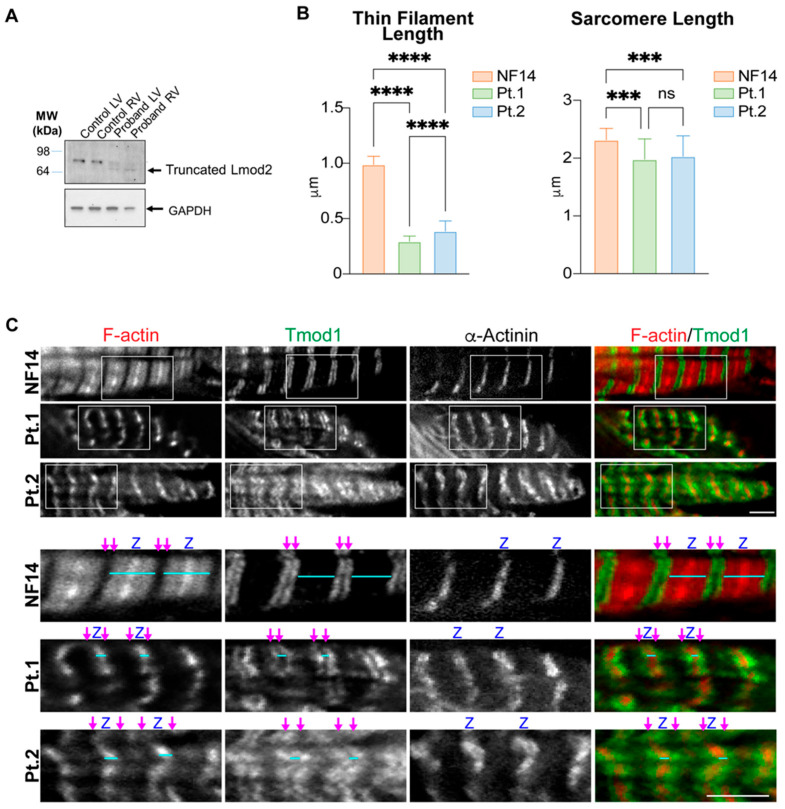
Pathologic Impact of *LMOD2* c.1193G>A (p.W398*) Variant. (**A**) Immunoblot of Lmod2 protein in proband explant left ventricle (LV) and right ventricle (RV) affected with homozygous *LMOD2* c.1193G>A (p.W398*) and in LV and RV in an age-matched control. Gapdh was used as a loading control. (**B**) Thin filament lengths and sarcomere lengths in LV measured by Tmod1 staining of the proband (Pt.2), a 14-month-old non-failing control (NF14), and the first case reported with the homozygous *LMOD2* c.1193G>A (p.W398*) variant (Pt.1). Measurements were obtained using the DDecon ImageJ plugin. Values are mean ± SD, and n = 44, 22, and 37 measurements (NF14, Pt.1, and Pt.2, respectively). *** *p* < 0.001 and **** *p* < 0.0001, one-way ANOVA. ns: non-significant (**C**) Representative immunofluorescence images of LV from NF14, Pt.1, and Pt.2. Fluorescently labeled phalloidin stains F-actin (red), Tmod1 staining marks pointed ends (green), and α-actinin marks Z-discs (Z). Magenta arrows mark thin filament pointed ends, and cyan lines show examples of thin filament arrays used for measurements. Scale bar: 2 µm.

**Table 1 cells-12-01455-t001:** Summary of LMOD2-Related Studies Reported to date.

Authors, Year	Organism, Sex, Number, Family History	Inheritance, Variant	Panel	ECG Features	Gross DCM Course	Outcome	Cardiomyocyte Histology (H&E, PAS)	IF or EM Findings	mRNA Levels	Lmod2 Protein on Immunoblot
Present case	Human male ×1, Mixtec Mexican	Homozygous c.1193G>A (p.W398*)	WES	Wide complex VTA	at 4 mo, triggered by infection	Cardiac transplant at 7 mo	Unremarkable	Regular sarcomere length	N/A	Absent (OriGene_AP52508PU-N)
Ahrens-Nicklas et al., 2019	Human female ×1, Mexican	Homozygous c.1193G>A (p.W398*)	WES	VT and ectopy	at birth	Cardiac transplant at 10 mo	Hypertrophy, vacuolation	Sarcomere disorganization, thin filament shortening	Pre-mRNA normal, mature mRNA decreased	Absent (Santa Cruz E13)
Greenway et al., 2021	Human females ×2, Vietnamese	Comp Hetc.1243_1244del (p.L415Vfs*108)c.1537C>T (p.R513*)	184-gene panel	atrial ectopic tachycardia (Sib 2)	at birth (Sib 1); suspected at 37 wk GA, confirmed at birth (Sib 2)	Therapeutic withdrawal at 24 do (Sib 1), 31 do (Sib 2)	Perinuclear halo, cytoplasmic vacuolation	Myofibril misalignment, broad Z-discs, SR dilation	(N/A)	(N/A)
Yuen et al., 2022	Human males ×2, Egyptian	Homozygous c.273+1G>A	WES	SVT (III:3)	at birth, full term (III:3) and 28 + 3 wks (III:4)	Death in 17 h (III:3) and 9 h (III:4)	(N/A)	Sarcomere disorganization, thin filament shortening	Absent pre-mRNA	Absent (Santa Cruz sc-135493, S12)
Lay et al., 2022	Human male ×1, Mexican	Homozygous c.1243_1244delCT (p.L415Vfs*108)	WES	SVT during 1st episode	at 9 mo following gastroenteritis; interim recovery; recurrence s/p volvulus surgery	Cardiac transplant at 14 mo	moderate hypertrophy	Sarcomere degeneration with streaming; actin filament length N/A	(N/A)	(N/A)
Pappas et al., 2015	Mouse, sex not provided	LacZ KO	(N/A)	(N/A)	Progressive	Juvenile death	(N/A)	Sarcomere disorganization, thin filament shortening	Absent	Absent
Li et al., 2016	Mouse, males and females	piggyBac KO	(N/A)	Long QTc	Progressive	Juvenile death	Unremarkable	Sarcomere disorganization, thin filament shortening	Absent	Absent

## Data Availability

The datasets used and analyzed during the current study are available from the corresponding author upon reasonable request. The study was registered in dbGaP under Novel Gene-Environment Regulatory Circuit in Chamber-Specific Growth of Perinatal Heart, Study ID: 45333. The stable dbGaP accession for this study is phs002725.v1. p1. Upon acceptance of this manuscript, WES data will be deposited to dbGaP and made publicly available prior to publication.

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
