# Peer review of "Whole-Exome Sequencing Identifies Homozygote Nonsense Variants in LMOD2 Gene Causing Infantile Dilated Cardiomyopathy"

_cells, 2023, doi:10.3390/cells12111455_

Round 1

Reviewer 1 Report

The authors reported  the fifth case of dilated cardiomyopathy related to biallelic variants in the LMOD2 gene. The phenotypic and histological features of this  c.1193G>A (p.W398*) nonsense variant are quite similar to what previously reported. The is well-conducted and well documented case report but its pertinence as a research article is not obvious.

Major comments:

-Results included an extensive discussion on the interpretation of candidate variants. This part is interesting but should mostly be move to the discussion part.

Author Response

WE thank the reviewer for positive comments.

We agree with the reviewer that the interpretation variant can be moved to discussion. We have moved them as recommended.

Given the article includes experimental data to validate the variants including sequencing, Immunohistochemistry, figures and and quantification, We argue that our article is qualified as an original article.

Reviewer 2 Report

The manuscript describes a further case of dilated cardiomyopathy due to homozygous variants in LMOD2 (c.1193G > A, p. Trp398*). This is the second report of this particular variant and the authors note interesting differences in disease onset (infantile rather than neonatal). This case report is very interesting and represents an important addition to the literature, further supporting that genetic testing for LMOD2 variants is indicated in patients with neonatal and infantile DCM.

Suggestions for improvements:

Minor:

-          Line 26: delete the in front of LMOD2

-          Line 74: LMOD2 should be italicized if the authors are referring to the gene here. Generally, this sentence is difficult to understand. Perhaps the authors mean “including variants in LMOD2” rather than “including LMOD2”.

-          LMOD2 abbreviation is defined in line 85

-          Figure 2 legend: (D) day X of admission should perhaps be changed to day 11 of admission?

-          Line 242: change to “echocardiogram demonstrated stabilisation following LVAD implantation” or similar.

Major

-          The manuscript is unnecessarily wordy with a large amount of information provided that is not necessarily relevant. Substantial edits to improve language clarity and streamline the content are required.   

-          The lengthy introduction about panel vs WES vs WGS is not relevant to this case report which predominantly describes a new variant in LMOD2 in a patient with infantile onset DCM. A more targeted/shorter introduction would be preferrable.  

-          LMOD3 is expressed at relatively constant levels throughout life in striated muscle (both cardiac and skeletal, see Yuen et al 2014). Thus, I strongly disagree with calling it “fetal” leiomodin. I am aware that LMOD3 is often referred to as fetal in various places (including Uniprot), however, in my opinion it is best not to further propagate this questionable label.

-          The author’s state: “LMOD2 has not been listed on any of the cardiomyopathy gene panels available thus far “. This information appears to be outdated and should be revised appropriately. For example, cardiomyopathy panel from BluePrintGenetics appears to include LMOD2 (https://blueprintgenetics.com/tests/panels/cardiology/dilated-cardiomyopathy-dcm-panel/?pdf).

-          Line 160: please supply catalogue number for both antibodies used. Suppliers are offering several antibodies for actinin and it is unclear which one the authors used. Also, in the case of LMOD2, the supplier may start offering several LMOD2 antibodies in the future.

-          Section 2.7 – thin filament measurements. What are the authors actually measuring here? No details are given beyond how images were prepared prior to measurements. Also, immunostaining for actinin only visualises the z-disc so only sarcomere length/z-disc structure can be assessed using actinin IHC (spacing between actinin bands). I am not sure how measuring sarcomere length is useful in the absence of thin filament length measurements. Sarcomere lengths are highly variable even in healthy tissue and depend on how much the tissue was stretched prior to fixation or how much it contracted due to calcium exposure of the sarcomere. It is not a reliable indicator for thin filament shortening. The authors also don’t compare to control tissue fixed in the same way and thus no conclusions can be drawn on whether their findings are different to healthy cardiac muscle. Statement in line 469-471 should be adjusted to reflect these considerations or the authors should consider removing this data altogether given the significant limitations. I suggest performing thin filament measurements following protocols described by C Gregorio’s group (University of Arizona). If this is technically challenging, seeking out an experienced collaborator my be helpful. Image J (Fiji) may be a good alternative software for image processing and analysis. Again, working with an experienced colleague or collaborator may be useful.

-          Line 206: the non-clinical audience will likely not know what G8P3 means. Generally, the clinical description is very specialised and only accessible to a cardiology specialised audience. This may not be appropriate for this journal.

-          A LVAD appears to overcome the clinical presentation of heart failure. Given that LMOD2 is critical in all parts of the heart, it would be appropriate to comment on right ventricular function. Was any abnormal function of the right heart noted during clinical review? Can the authors comment/hypothesis why the right heart is seemingly unaffected?

One possible hypothesis is that the right heart is subjected to less stretch than the left heart due to the higher load in the systemic circulation compared to the pulmonary circulation. The concept that increased load in the systemic circulation at birth is causing the onset of disease in LMOD2-related cardiomyopathy is discussed in Yuen et al 2022: “After birth, the heart undergoes a dramatic change in hemodynamics, including an increase in systemic vascular resistance. We hypothesize, that the resulting increase in cardiac afterload/preload stretches the cardiac ventricles, leading to longer end-diastolic sarcomere lengths. Since the thin filaments are shorter, the thin/thick filaments overlap is reduced or absent at long sarcomere lengths resulting in reduced contractile force and ejection fraction (also discussed in 8).”

Is it perhaps possible that delayed disease onset in the presented proband is due to structural abnormalities of the heart at birth decreasing the load/stretch of the left ventricle at birth? I think the authors are trying to discuss the concept that fetal circulation may be protective. However, the writing needs to be significantly improved to make the content clearer.

-          Line 265-268 may be better placed at beginning of section 3.1, where family history was also briefly mentioned. Was abnormal cardiac function definitively excluded for the 3 spontaneous abortions and 2 fetal demises? The statement “family history was negative for children or infants with heart defects, arrhythmias, or sudden death” suggests that previous 5 deaths are not heart related. Can the authors please clarify?

-          The authors provide a detailed pathogenicity interpretation for other variants identified warranting their exclusion as disease causing in section 3.4.3. While it is important that these variants were closely assessed, for this case report it is probably sufficient to state that all three variants were excluded based on evaluation of phenotypic concordance, population frequency, etc.

-          Western blot in Figure 6A shows two weak bands in the proband which could be LMOD2 protein: one at the size of full length LMOD2 and one below. Given the late onset cardiomyopathy, clear evidence excluding or confirming residual LMOD2 expression should be provided. What is the predicted size of truncated LMOD2 resulting from the reported variant? It may be helpful to point out the expected location of the truncated product on the Western blot so the reader can easily confirm that no protein product is present. Also, is it possible that LMOD2 is differently spliced due to this variant?

-          I am not sure the authors correctly interpreted the statement in Yuen et al 2022 regarding LMOD3 as a protein potentially compensating for loss of LMOD2. Yuen et al did not state that LMOD3 is “an unlikely candidate” for compensation. Yuen et al 2022 stated that there is no evidence that LMOD2 and LMOD3 have overlapping functions. This does not mean LMOD3 is definitely unable to modify disease arising from LMOD2 loss. In fact, it would be interesting to investigate LMOD3 levels in the present proband and the proband reported by Ahrens-Nicklas to determine whether LMOD3 protein is increased in either patient in response to LMOD2 loss.

Figure 6D shows LMOD2 RNA expression in mice. No details on mouse ethics and how this experiment was performed are given as far as I can tell. The result is only mentioned in the discussion and not in the results section. No statistical analysis has been performed. While the thought is interesting, I think this result is very preliminary and further studies should be performed to investigate this in more detail before publishing.

Author Response

We thank Reviewer 3 for valuable comments.

We have summarized our responses in the attached file.

Marlin 

Reviewer 3 Report

I’ve read with interest the manuscript by Sono et al., which describes a case of infantile-onset dilated cardiomyopathy due to homozygous LMOD2 truncating mutation. There are still few reports on LMOD2-associated neonatal DCM and this is the first case with relatively late infantile disease-onset, thus this manuscript gives relevant insight into clinical presentation and disease knowledge. I totally agree with the need to perform first-tier WES or WGS in neonatal/infantile cardiomyopathies. The work is comprehensive and well-conducted and I believe that should merits publication as it is.

Author Response

We thank the reviewer for positive comments and recommendation.

Round 2

Reviewer 1 Report

The manuscript has been significantly improved. There are still some major concerns: For instance, the methodologies used to quantify the thin filament length must be devoid of biais.

Fig. 6. Length of thin filaments were measured on immunostaining images. However, the pictures shown in the manuscript are blurred. Could you improve image quality and provide focused images? This is of critical importance to determine thin filament length and support conclusions. As it, it was not possible to be confident on filament length.

Author Response

Dear Reviewer,

We highly appreciate your input. 

We highly appreciate the recommendation by the reviewer. To address this comment, we collaborated with Carol Gregorio’s laboratory who has > 25 years of experience measuring thin filament lengths (e.g., Ahrens-Nicklas et al., 2019 (8). As such, we have replaced the panels with images her laboratory generated and quantified.

Ahrens-Nicklas RC, Pappas CT, Farman GP, Mayfield RM, Larrinaga TM, Medne l, Ritter A, Krantz ID, Murali C, Lin KY, Berger JH, Yum SW, Carreon CK, Gregorio CC. 2019. Disruption of cardiac thin filament assembly arising from a mutation in lmod2: a novel mechanism of neonatal dilated cardiomyopathy. Sci Adv. 2019 5(9):eaax2066. PMID:31517052.

Please refer to Figure 6.B and C of the current revised manuscript.

Best,

Marlin Touma, MD, PhD

UCLA

Reviewer 2 Report

Line 73: Consider referring to Lmod3 as “striated muscle leiomodin”.

It may be best to remove “early after birth” in line 77 as, to the best of my knowledge, thin filaments reach their mature length prenatally or around the time of birth.

Line 81: This sentence needs to be revised. “… inconsistent indications ?for testing?...” or “… inconsistent indications for (eg. (familiar dilated cardiomyopathy, etc...)”

Line 242: please state supplier of Gapdh antibody.

Section 2.7. as per previous comments, the authors do not state how thin filament lengths were measured. The information given is not sufficient for the reader/reviewer to evaluate the quality of the data. For example, how many thin filaments were measured in each image. From where to where exactly was the measurement taken. Which software was used for measurements of thin filament lengths? Thin filament lengths have been shown to vary depending on sarcomere lengths (slightly) so it is important to confirm the sarcomere lengths were similar in sample/control and between areas.

Generally, the authors do not appear to have the required expertise in house to perform this experiment and my reservations concerning the immunofluorescence presented in revised figure 6 remain despite the revision. You cannot simply count sarcomere units and infer thin filament lengths. The analysis performed by the authors is flawed. The tissue has to be thawed in relax solution, stretched and fixed prior to analysis. This would allow resolving the single Tmod band observed in the presented images into doublets presenting the thin filament pointed ends. My recommendation remains the same as in my previous review: collaborate with an expert or remove this data from the manuscript.

Line 286 should be combined with info in line 309-311 and 362-363. Keeping all the family history in one location will make this section easier to comprehend.

Line 627-631: I don’t think the authors have sufficient evidence to support the conclusions drawn here. Knock out tissue would be required to confirm that the top band is indeed non-specific. Also, the lower band may represent truncated Lmod2 (since it migrates at the expected molecular weight and is absent from control tissue), however, the band has not been unambiguously been identified as Lmod2. The statements in this section should be adjusted to correctly represent the evidence provided.

Line 740-763: This paragraph needs to be revised ideally by a proficient English speaker/writer. In its current form it does not correctly discuss statements made by Yuen et al 2022.

Section 2.8: All RNAseq results were removed so this methods section is no longer required?

Confirm that all abbreviations are defined when they are first mentioned and not again defined later (eg WGS).

As per my previous recommendation, the description of excluded variants is excessive and should be limited to state clear points that led to the exclusion of each variant only. It is not typical for articles of this kind to describe excluded variants in this amount of detail. Alternatively, this information could be moved to supplements in a table format.  

Reviewer 2 asks the question “Is it perhaps possible that delayed disease onset in the presented proband is due to structural abnormalities of the heart at birth decreasing the load/stretch of the left ventricle at birth?”
Response. While this hypothesis worths investigations, we do not have evidence of underlying structural defect in our proband that may affect the flow and ventricular stretch.

The authors write the following in the manuscript: “bidirectional shunt through a pa- 292
tent ductus arteriosus and a patent foramen ovale (PFO) versus small secundum ASD 293
(ssASD). The tricuspid and mitral valves showed mild insufficiency.

I am not a cardiologist but are the authors not describing structural heart abnormalities here?

Author Response

Dear reviewer,

We deeply appreciated your comments and suggestion and that helped us to make substantial revisions that helped to improve the manuscript significantly. We have summarized our responses in attached Pdf file. in addition to the revised  track-changed manuscript.

Marlin Touma

We look forward to your valuable input,

Round 3

Reviewer 2 Report

I note that the authors have substantially improved this manuscript and all my concerns have been addressed. I believe collaborating with Prof Gregorio and comparing thin filament lengths between the both probands with identical LMOD2 variants has lifted the scientific impact of this article.

This paper will be a valuable addition to the published literature .

Minor edits:

A few more typographical/language issues remain to be addressed throughout the manuscript.

In particular, newly inserted text 491-503 could benefit from editorial review to streamline the content and improve written presentation.

Line 599: Yuen et al hypothesized that this disease does not manifest before? birth....

Author Response

Line 491-503 should stay as they are. Line 599 fixed